# Er:YAG Laser Alleviates Inflammaging in Diabetes-Associated Periodontitis via Activation CTBP1-AS2/miR-155/SIRT1 Axis

**DOI:** 10.3390/ijms25042116

**Published:** 2024-02-09

**Authors:** Min Yee Ng, Cheng-Chia Yu, Szu-Han Chen, Yi-Wen Liao, Taichen Lin

**Affiliations:** 1School of Dentistry, Chung Shan Medical University, Taichung 40201, Taiwan; ngminyee_92@hotmail.com (M.Y.N.); ccyu@csmu.edu.tw (C.-C.Y.); jasminne1117@gmail.com (S.-H.C.); 2Department of Dentistry, Chung Shan Medical University Hospital, Taichung 40201, Taiwan; 3Institute of Oral Sciences, Chung Shan Medical University, Taichung 40201, Taiwan; rabbity0225@gmail.com; 4Department of Medical Research, Chung Shan Medical University Hospital, Taichung 40201, Taiwan

**Keywords:** diabetic periodontitis, Er:YAG laser, miR-155

## Abstract

Periodontitis is a significant health concern for individuals with diabetes mellitus (DM), characterized by inflammation and periodontium loss. Hyperglycaemia in DM exacerbates susceptibility to periodontitis by inducing inflammaging in the host immune system. The use of erbium-doped yttrium–aluminum–garnet laser (ErL) in periodontitis treatment has gained attention, but its impact on diabetic-associated periodontitis (DP) and underlying mechanisms remain unclear. In this study, we simulated DP by exposing human periodontal ligament fibroblasts (PDLFs) to advanced glycation end products (AGEs) and lipopolysaccharides from *P. gingivalis* (Pg-LPS). Subsequently, we evaluated the impact of ErL on the cells’ wound healing and assessed their inflammaging markers. ErL treatment promoted wound healing and suppressed inflammaging activities, including cell senescence, IL-6 secretion, and p65 phosphorylation. Moreover, the laser-targeted cells were observed to have upregulated expression of CTBP1-AS2, which, when overexpressed, enhanced wound healing ability and repressed inflammaging. Moreover, bioinformatic analysis revealed that CTBP1-AS2 acted as a sponge for miR155 and upregulated SIRT1. In conclusion, ErL demonstrated the ability to improve wound healing and mitigate inflammaging in diabetic periodontal tissue through the CTBP1-AS2/miR-155/SIRT1 axis. Targeting this axis could represent a promising therapeutic approach for preventing periodontitis in individuals with DM.

## 1. Introduction

Periodontitis is a localized inflammation of the tooth-supporting tissues and can lead to the destruction of underlying connective tissues and alveolar bone [1]. This local disease has a bidirectional relationship with diabetes mellitus (DM) and it has been demonstrated that periodontal management has a considerable impact on glycemic health in DM patients and vice versa [2,3,4]. Periodontitis often arises due to the host’s immune response to microbial triggers present in the dental plaque, while the accumulation of advanced glycation end product (AGE) levels in DM magnifies this host response [5]. AGEs are acknowledged for inducing inflammaging, a state constituted by low-grade chronic inflammation and pre-aging in various periodontal tissues including human gingival fibroblast, keratinocytes, and macrophages [6,7,8,9,10]. Inflammaging was believed to accelerate the progression of DP through several mechanisms, including cellular senescence and senescence-associated secretory phenotype (SASP) [11,12,13,14,15].

Cellular senescence is defined as a permanent cell cycle arrest in response to cellular damage or stress. For instance, AGEs in DM interact with their receptors (RAGEs) in cells and result in prolonged endoplasmic reticulum (ER) stress, leading to senescence or premature aging [16]. These ‘aged’ cells can exhibit senescence-associated secretory phenotype (SASP) by generating various pro-inflammatory cytokines including interleukin (IL)-6 [14,15,17,18]. Elevated activity of cell senescence and their SASP can create a detrimental pro-inflammatory milieu, which gives rise to tissue dysfunction and wound healing insufficiency [19]. Prolonged inflammation can provoke DNA damage, which in turn can aggravate senescence and perpetuate this vicious cycle [20]. Thus, it is crucial to address these in oral cells in order to limit tissue destruction and treat periodontitis. 

In light of that, the erbium-doped yttrium–aluminum–garnet laser (ErL) has been widely known for its application in the treatment of periodontal diseases [21,22,23,24]. It is a solid-state crystal laser that operates in the infrared spectrum at 2940 nm. It has been shown to successfully remove dental calculus and decontaminate infected root surfaces [25,26]. While a few studies have revealed that ErL therapy benefited DM patients with periodontitis in terms of their periodontal parameters [27] and glycaemic control [28], little is known regarding its molecular mechanism in targeting inflammaging in DP. 

Interestingly, it was found that people with DM have reduced expression of CTBP1-AS2 in their peripheral blood mononuclear cells [29]. This long non-coding RNA CTBP1, known as C-terminal-binding protein 1, is a transcriptional corepressor that is crucial for the regulation of gene expression and subsequent biological processes, including cell proliferation, differentiation, and apoptosis [30]. Given that the upregulation of CTBP1-AS2 expression was able to suppress high glucose-induced inflammation in diabetic nephropathy in an in vitro study [31], it would be valuable to look into whether ErL targets inflammaging in DP via this long non-coding RNA, as well as its interplay with other microRNAs. 

Periodontal ligament fibroblasts (PDLFs), which are found between the root cementum and alveolar bone, play a pivotal role in the maintenance of periodontal health. Their responsibilities include synthesizing extracellular matrix proteins and regulating occlusal force [32]. Notably, they actively contribute to local immune responses and facilitate tissue regeneration [33,34,35] and cellular defects in them pose threats to the integrity of periodontal tissue. Consequently, we sought to explore the influence of ErL on inflammaging in DP-simulated PDLFs and its underlying molecular mechanism revolving around the CTBP1-AS2 and its target genes. The null hypothesis states that ERL does not exert a significant impact on inflammaging in DP-simulated PDLFs, and it does not modulate the expression of CTBP1-AS2 and its target genes.

## 2. Results

### 2.1. Advanced Glycation End Products (AGEs) Reduce the Cell Proliferation in PDLFs

First of all, the effects of various concentrations of AGEs on the cell viability were assessed to determine the dose of AGEs for subsequent experiments. AGEs significantly inhibited cell proliferation in a dose-dependent fashion (Figure 1). The lowest effective dose of AGEs, 62.5 μg/mL, was used for subsequent experiment.

### 2.2. AGEs/LPS-Stimulated PDLFs Impair Wound Healing While ErL Irradiation Reverses the Phenomena

To mimic diabetic periodontitis in vitro, PDLFs were cultured with AGEs and LPS. PDL-1 and PDLF-2 refer to PDLFs obtained from two different individuals. As shown in Figure 2, wound healing was shown the most impaired in PDLFs stimulated with both LPS and AGEs, in comparison to control (no stimuli) or AGEs or LPS stimuli alone. When ErL was given as an intervention, an energy density of 3.6 J/cm^2^ did not have significant improvement while the higher energy densities, 4.2 and 6.3 J/cm^2^, increased the wound healing capability.

### 2.3. ErL Irradiation Targets the Upregulated Inflammaging Activities in PDLFs Subjected to AGEs/LPS 

Given that inflammaging’s markers include cellular senescence along its senescence-associated secretory phenotype (SASP) [11,12,13,14,15], our assessment focused on the impacts of AGEs/LPS and ErL on these parameters. As expected, AGE/LPS stimulus markedly increased inflammaging, characterized by the upregulated cell senescence expression (Figure 3A), IL-6 secretion (Figure 3B), and NF-κB signaling (Figure 3C). Meanwhile, ErL irradiation led to a significant inhibition of cell senescence activity (Figure 3A), IL-6 secretion (Figure 3B), and NF-κB signaling (Figure 3C). Collectively, these findings demonstrated that ErL irradiation has an anti-inflammaging trait in DP in vitro.

### 2.4. The Stimulatory Effects of ErL Irradiation on CTBP1-AS2 and the Overexpression of CTBP1-AS2 Represses AGE/LPS-Induced Poor Wound Healing and Inflammaging

Considering that DM patients exhibited reduced expression of CTBP1-AS2 in their peripheral blood mononuclear cells [29], we investigated whether the expression of this lncRNA was similarly affected in PDLFs subjected to AGEs/LPS. Interestingly, the expression of CTBP1-AS2 in these DP-simulated cells was found tremendously lower compared to that of the control group with no AGE/LPS stimulus. However, following intervention with laser irradiation, this downregulated lncRNA was restored to a level significantly higher than that of the control group (Figure 4A). Given that Er:YAG laser (ErL) was effective in ameliorating inflammaging and increasing the expression of CTBP1-AS2, it was crucial to explore the significance of this lncRNA in the suppression of inflammaging. Subsequently, we overexpressed CTBP1-AS2 in cells and assessed wound healing, cell senescence, and ROS production. It was demonstrated that the CTBP1-AS2 overexpression reversed all AGE/LPS-induced effects including poor wound healing (Figure 4B), enhanced cell senescence (Figure 4C), and ROS production (Figure 4D).

### 2.5. miR-155 Is a Direct Target of CTBP1-AS2 and Inhibition of miR-155 Downregulates Inflammaging

CTBP1-As2 has been shown to translocate into the cytoplasm to exert its influence on target molecules [36]. Here, we revealed that the level of CTBP1-AS2 expression was markedly upregulated in the cytoplasm of PDLFs after ErL irradiation (Figure 5A). Previous studies have demonstrated that CTBP1-AS2 functions as a sponge for miR-155 in human glomerular mesangial cells, and there is notable upregulation of miR-155-5p in the peripheral blood of patients with diabetic nephropathy [31]. Given this, our investigation aimed to assess the potential interaction between CTBP1-AS2 and miR-155 in PDLFs. By using bioinformatics analysis, a putative miR-155 binding site on CTBP1-AS2 was discovered and the complementarity between the 3′ UTR regions of CTBP1-AS2 and miR-155 was illustrated (Figure 5B). When PDLFs were transfected with CTBP1-AS2, the mRNA level of miR-155 was decreased tremendously (Figure 5C). Reporter plasmids containing either full-length (Wt-) or mutated (mut-CTBP1-AS2) forms of the miR-155-binding region were constructed and cotransfected with miR-155 mimics into PDLFs. The luciferase activity of Wt-CTBP1-AS2 vector was reduced when cotransfected with miR-155 mimics, whereas no significant change was present in the mut-CTBP1-AS2 vector in PDLFs (Figure 5D). This result demonstrated there is a direct interaction between CTBP1-AS2 and miR-155. When miR-155 inhibitor was introduced, the inflammaging expression in PDLFs was reduced, as defined by the decrease in cell senescence (Figure 5E) and ROS levels (Figure 5F). This showed that CTBP1-AS2 alleviated inflammaging via directly binding to miR-155 and reducing its expression.

### 2.6. CTBP1-AS2 Ameliorates Inflammaging in PDLFs via Regulation of miR155/SIRT1 Signaling

miR-155 has been identified as a direct repressor of the histone deacetylase Sirtuin 1 (*SIRT1*) in various pathologies including diabetic nephropathy [37,38] and diabetic osteoporosis in vitro [39]. However, the study on this relationship in diabetic periodontitis is limited and therefore, we examined whether miR-155 interacted with *SIRT1* in PDLFs. We employed the luciferase reporter assay to examine the promoter activity and the alignment of the *SIRT1* 3′ UTR region with miR-155 is shown in (Figure 6A). Cells cotransfected with miR-155 mimic and reporter plasmid containing wild-type *SIRT1* 3′ UTR displayed reduced luciferase activity, whereas the luciferase activity was unaffected by cotransfection of miR-155 and reporter vector cloned with *SIRT1* 3′ UTR containing a mutated binding sequence (Figure 6B). When miR-155 mimics were added to PDLFs, the mRNA level of *SIRT1* decreased significantly (Figure 6C). These data demonstrated that miR-155 negatively regulated *SIRT1* gene expression. To validate if CTBP1-AS2 targets *SIRT1*, we compared the cell senescence in CTBP1-AS2-silenced PDLFs with and without *SIRT1* mimic. Our results showed that the increased cell senescence of CTBP1-AS2-silenced PDLFs was downregulated when *SIRT1* mimics were added (Figure 6D). Taken together, these results showed that CTBP1-AS2 possesses the ability to promote *SIRT1* expression through the inhibition of miR-155 in ameliorating inflammaging.

## 3. Discussion

In the current investigation, we present the first demonstration illustrating the protective role of CTBP1-AS2 in safeguarding PDLFs against AGE/LPS stimuli by the attenuation of inflammaging through the regulation of the miR-155/*SIRT1* axis. These findings shed light on the underlying mechanism through which ErL intervention addresses the complexities of diabetic periodontitis (DP).

Upon examining the impact of AGEs and LPS from *P. gingivalis* on PDLFs, there was a synergistic and detrimental effect on the cell’s wound healing, aligning with findings from previous studies [40,41]. When these cells were intervened with laser treatment, particularly at higher energy densities, they demonstrated a greater benefit in the impaired wound healing efficacy, comparable to the study of Lin et al. [42]. In addition, the positive effect of ErL on wound healing has been observed in various cells, including gingival fibroblast and osteoblast [43,44,45]. Though not explored here, it was thought that the laser’s enhancement of wound healing may be attributed to augmented cell proliferation and migration [42,46]. Given that poor wound healing is associated with inflammaging [47], it comes as anticipated in this study that AGE/LPS-stimulated PDLFs exhibited signs of inflammaging. This was evidenced by a significant surge in cell senescence, IL-6, and NF-κB signaling activation [15,17,48,49]. The NF-κB pathway, recognized as a pivotal regulator in inflammaging, plays a crucial role in the release of inflammatory cytokines and cellular senescence, along with SASP [50,51,52,53,54]. In contrast, ErL intervention dramatically reversed the induced inflammatory phenomena. Notably, ErL as adjunct therapy was revealed to reduce IL-6 expression in gingival crevicular fluid (GCF) in patients with chronic periodontitis compared to scaling and root planing (SRP) alone [55]. 

Numerous studies have uncovered the role of long non-coding RNA (lncRNA) dysregulation in periodontitis pathogenesis, orchestrating diverse regulatory mechanisms. Accumulating evidence suggests that their influence encompasses critical processes in periodontal cells, such as osteogenic differentiation, inflammation, proliferation, apoptosis, and autophagy [56]. Interestingly, AGE/LPS stimuli resulted in a diminished expression of lncRNA CTBP1-AS2 in PDLFs. The level of this lncRNA was significantly restored post-ErL intervention, surpassing levels observed in the control group. This lncRNA emerges as a crucial player in mitigating inflammaging, as its overexpression successfully reversed all AGE/LPS-induced consequences in wound healing, cell senescence, and ROS production. Given that DM patients have been reported to have lower levels of CTBP1-AS2 in their peripheral blood mononuclear cells [29] and replenishing CTBP1-AS2 expression can prevent high glucose-induced diabetic nephropathy [31], the prospect of employing this lncRNA as a targeted therapeutic approach or even a diagnostic marker in individuals with DP holds considerable promise. 

The current study reveals that CTBP1-AS2 ameliorated inflammaging in PDLFs via regulation of miR155/*SIRT1* signaling. Similar to Wang’s study [31], CTBP1-AS2 was observed to act as a sponge for miR-155. miR-155, previously found to be associated with the DNA damage response, has been linked to the buildup of cell senescence, SASP, and ultimately inflammaging [57]. In various DM-associated complications such as diabetic nephropathy and diabetic osteoporosis in vitro, the high levels of miR-155 have been recognized as a direct suppressor of histone deacetylase *SIRT1* [37,38,39]. *SIRT1*, a nicotinamide adenosine dinucleotide (NAD)-dependent deacetylase and a class III histone deacetylase [58], is known to contribute to osteoblastic differentiation and alleviation of inflammation in periodontal ligament cells [59,60,61]. 

One limitation of this study is that it is solely an in vitro investigation, and thus, the findings may not fully capture the complexities of DP in vivo. Further animal and pre-clinical studies would be necessary to correlate periodontal tissue loss severity, glycemic control in DM, and the studied lncRNA expression. Despite this limitation, the insights gained from this study hold significant promise, enhancing our understanding of inflammaging in DP through the CTBP1-AS2 pathway. Utilizing this lncRNA as a targeted therapeutic approach could involve strategies to restore or enhance its expression, aiming to mitigate the progression of periodontal complications in individuals with DM. Additionally, its potential role as a diagnostic marker could offer a non-invasive means of assessing the risk or severity of DP in DM patients, enabling earlier intervention and personalized treatment strategies. Furthermore, considering the increasing association of inflammaging with various inflammatory diseases, this knowledge holds the potential to advance therapeutic strategies, addressing heightened host responses not only in DP but also in other DM-related complications, including diabetic nephropathy. 

Collectively, these findings highlight the role of CTBP1-AS2 in enhancing *SIRT1* expression by inhibiting miR-155 in its amelioration of inflammaging. As illustrated in Figure 7, the diminished expression of CTBP1-AS2 in DP tissues leads to the upregulation of miR-155, subsequently suppressing *SIRT1* expression, thereby contributing to inflammaging. The irradiation of ErL, on the other hand, effectively restored the impaired levels of this lncRNA and reversed inflammaging.

## 4. Materials and Methods

### 4.1. Tissue Collection, Cell Culture, and Reagents

All steps were carried out in line with Chung Shan Medical University Hospital’s Institutional Review Board-approved norms (IRB approval number: CSMUH No: CS18196). After consent was obtained, PDLFs were isolated from two healthy individuals who had their premolars extracted for orthodontic purposes. From the middle one-third of the root, PDLFs were extracted and Dulbecco’s modified Eagle’s medium containing 10% fetal bovine serum, 100 units/mL penicillin, and 100 mg/mL streptomycin were used to keep the cells alive.

### 4.2. Laser Irradiation

ErL (Erwin AdvErlTM, wavelength 2940 nm, pulse width 250 μs, J. Morita Mfg, Kyoto, Japan) with a 2940 nm emission wavelength was used. Before the irradiation, the medium of the cells was first removed to expose the monolayer to ErL. With no covering sleeve or contact point for the handpiece, the laser was pointed perpendicularly at the culture dish from a height of 15 or 20 cm. The energy densities were determined at 3.6, 4.2, and 6.3 J/cm^2^ by manipulating the laser parameters, which are based on the previous studies [42]. Thermal-hypersensitive paper was used to mark the laser irradiation area to ensure the irradiation area fully covered the 35 mm cell culture dishes.

### 4.3. Cell Proliferation Assay 

Cells were seeded at 1 × 10^4^ cells/well in the 96-well plates with varied AGE concentrations at 37 °C and followed by laser irradiation at the 24th hour. After irradiation, the cells were further cultured at 37 °C for 48 h. Following that, a solution of 3-(4,5-Dimethylthiazol-2-yl)-2,5-diphenyltetrazolium bromide (MTT) was supplemented to each well and cultured for 3 h. Following that, the MTT formazan was dissolved in DMSO and measured spectrophotometrically at 570 nm. The optical density readings for each group were expressed as a percentage of the control.

### 4.4. Flow Cytometry Analysis

Using flow cytometry analysis, ROS production was examined by measuring the fluorescence strength of 2′,7′-dichlorofluorescein (DCF) in PDLFs stimulated with AGEs, LPS, and different intensities of ErL. The fluorescence of 2′,7′-dichlorofluorescein (DCF) as well as ethidium (ETH), are the products of oxidation of 2′,7′-dichlorodihydrofluorescein diacetate (DCFH-DA; Sigma–Aldrich, Madrid, Spain) and dihydroethidium (DHE; Molecular Probes, Eugene, OR, USA) with a sensitivity for H_2_O_2_/NO-based radicals and O-2, respectively. For 60 min and at 37 °C, the irradiated cells were incubated with 10 μM DCFH-DA or DHE preceding being washed twice with PBS. Flow cytometry (Becton–Dickinson, San Diego, CA, USA) was utilized to examine ETH fluorescence and DCF fluorescence of 10,000 cells at excitation and emission wavelengths of 488 and 525 nm, respectively.

### 4.5. Wound Healing Assay

Once seeded into a 12-well culture dish, the cells were cultivated till they reached up to 80% confluence. This was followed by creating a wound among the monolayer by scratching across the center of the well with a sterile 200 L pipette tip. Cells were allowed to develop for another 48 h before staining with crystal violet. At 0 and 48 h, the displacement of cells towards the wound site was measured using a microscope.

### 4.6. Senescence-Associated Beta-Galactosidase Staining

A senescence detection kit (BioVision, San Francisco, CA, USA) was utilized according to the manufacturer’s procedure to assess the SA-βGal-positive cell ratios. Irradiated cultures of 4 × 10^3^ cells were first washed with 1 mL PBS once and then fixed for 10 min at room temperature with the kit Fixative Solution. The cells were then stained with 470 µL of kit Staining Solution, 5 µL of Staining Supplement, and 25 µL of 20 mg/mL X-gal in DMF. This was followed by washing the samples twice using 1 mL of PBS. After that, each well received an additional 0.5 cc of the kit’s Staining Solution Mix and incubated overnight at 37 °C. The cells were then examined under a phase contrast microscope (200 total magnification) for the formation of blue color.

### 4.7. Western Blotting

The Western blot test was carried out in accordance with the procedure previously described [62]. Antibodies to phospho-p65 markers (p-p65) were employed as primary antibodies. Primary antibodies used included antibodies to phospho-p65 markers (1:500; cat. no. sc-134306; mouse monoclonal) from Santa Cruz Biotechnology, Dallas, TX, USA, and anti-GAPDH (Millipore, Bedford, MD, USA) was used as the loading control. Following blocking, the membranes were incubated with indicated primary antibodies followed by corresponding secondary antibodies. The immunoreactive bands were generated with an ECL-plus chemiluminescence substrate (Perkin-Elmer, Waltham, MA, USA) and taken with ImageQuant LAS 4000 Mini (GE Healthcare, Piscataway, NJ, USA). Each densitometric value was expressed as the mean ± standard deviation.

### 4.8. ELISA Analysis

An Il-6 ELISA kit (Detection Range: 7.8–2500 pg/mL; Sensitivity: <2 pg/mL) was used in the present study (Thermo Fisher Scientific, Waltham, MA, USA). IL-6 concentrations were quantified with an ELISA filter on a microplate reader with a 450 nm filter (MRX; Dynatech Laboratories, Chantilly, VA, USA) and each individual model was evaluated three times.

### 4.9. Cell Transfection

We constructed pcDNA3.1-CTBP1-AS2 (CTBP1-AS2) and pcDNA3.1 empty vector (Vector). For cell transfection, PDLFs in the logarithmic phase were transfected with these plasmids using Lipofectamine 2000 (Invitrogen, Carlsbad, CA, USA) prior to 48 h treatment with LPS and AGEs. Wound healing, cell senescence, and ROS production were then measured in the AGE+LPS-induced cells.

### 4.10. MiRNA-Targeting Gene Prediction and Dual-Luciferase Reporter Assay

The wild-type CTBP1-AS2-3′UTR was cloned into the β-gal control plasmid according to the manufacturer’s protocol. The mutant reporter was generated by replacing the original sequence ACGUUUU in the wild-type reporter with GCUAAUU. The ß-galactosidase activity of vector alone plasmid, the wild-type reporter, and the mutant reporter were normalized using the luciferase activity of a cotransfected plasmid expressing luciferase in order to represent background reporter activity. The reporter plasmid and miR-155 mimic or miR-Scramble were cotransfected into cells using a Lipofectamine 2000 reagent (LF2000, Invitrogen, Carlsbad, CA, USA). Firefly luciferase activity after normalizing to transfection efficiency represented reporter activity.

### 4.11. Lentiviral-Mediated RNAi for Silencing TGIF2

The pLV-RNAi vector was purchased from Biosettia Inc. (San Diego, CA, USA). The method of cloning the double-stranded shRNA sequence followed the manufacturer’s protocol. The oligonucleotide sequence of lentiviral vectors expressing shRNA that targets human TGIF2 was synthesized and cloned into pLVRNAi to generate a lentiviral expression vector. The target sequences for TGIF2 are listed as follows: Sh-TGIF2-1 5′-AAAA-GCTGCCAAATTCAGTCCTATTGGATCCAATAGGACTGAATTTGGCAGC-3′. Lentivirus production was performed by cotransfection of plasmid DNA mixture with lentivector plus helper plasmids (VSVG and Gag-Pol) into 293T cells (American Type Culture Collection, Manassas, VA, USA) using Lipofectamine 2000 (LF2000, Invitrogen, Carlsbad, CA, USA) [63].

### 4.12. RNA Isolation and Quantitative Reverse Transcription PCR (qRT-PCR)

Total RNA was isolated using TRIzol reagent (Invitrogen), and reversely transcribed to cDNA using the PrimeScriptcDNA Synthesis kit (Takara, Dalian, China), according to the manufacturer’s instructions. The synthesized cDNA was then used for qRT-PCR analysis using the SYBR Green Kit (Takara). The transcript levels of target genes were calculated according to the 2−ΔΔCT method using U6 or GAPDH as the internal reference. The specific primers used were as follows: CTBP1-AS2 (Forward primer) 5′-CGTTCTGATTCCTGGCATGG-3′, (Reverse primer) 5′-TACCTCATCGACGTTCCCAG-3. GAPDH (Forward primer) 5′-TGCACCACCAACTGCTTAGC-3′, (Reverse primer) 5′-GGCATGGACTGTGGTCATGAG-3′. miR-155 (Forward primer) GTTAATGCTAATTGTGATAGGGG, (Reverse primer) CATCATACCCTGTTAATGCTAAC. U6 (Forward primer) 5′-GAGGGTTAATGCTAATCGTGATAGG-3′, (Reverse primer) 5′-GCACAGAATCAACACGACTCACTAT-3′.

### 4.13. Statistical Analysis

Statistical Package of Social Sciences software (version 13.0) (SPSS, Inc., Chicago, IL, USA) was used for statistical analysis. Data from at least triplicate analysis were shown as mean ± SEM. Study variables were tested for normal distribution using the Shapiro–Wilk test. Intergroup comparisons of means in each parametric and non-parametric parameter were performed using the two-sample t-test and Wilcoxon rank sum test, respectively. Statistical significance was set at a two-tailed *p*-value of <0.05 for all tests.

## 5. Conclusions

In conclusion, ErL demonstrated anti-inflammaging influence in diabetes-associated periodontitis in vitro and it is proposed to be mediated through the CTBP1-AS2/miR-155/*SIRT1* axis and, therefore, could be a potential therapeutic target in DM patients with periodontitis. 

## Figures and Tables

**Figure 1 ijms-25-02116-f001:**
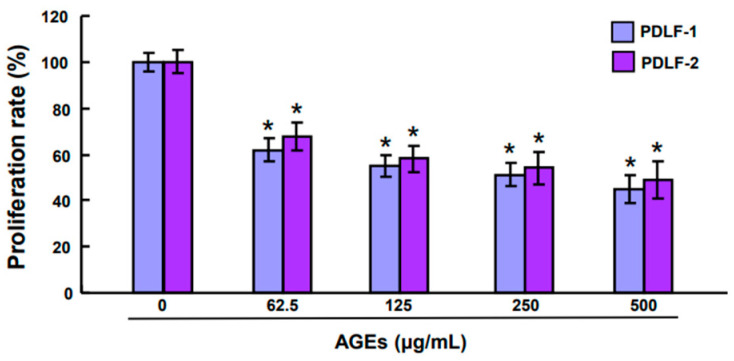
AGEs at 62.5μg/mL had a significant inhibition on the cell proliferation in PDLFs and its higher dosage further increased the suppression. Data were expressed in mean ± standard deviation. * *p* < 0.05 compared to control group.

**Figure 2 ijms-25-02116-f002:**
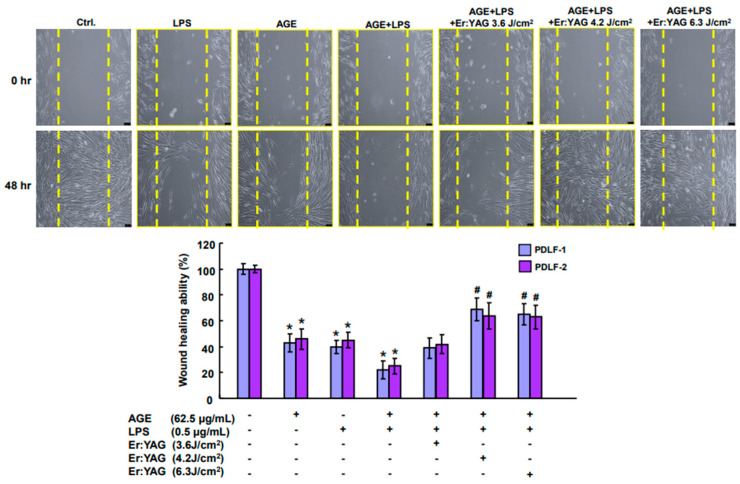
AGE/LPS-induced poor wound healing was reversed in PDLFs treated with ErL irradiation of higher energy densities. Data were expressed in mean ± standard deviation. * *p* < 0.05 compared to control group; # *p* < 0.05 compared to AGE/LPS only group. Scale bar, 100 µm.

**Figure 3 ijms-25-02116-f003:**
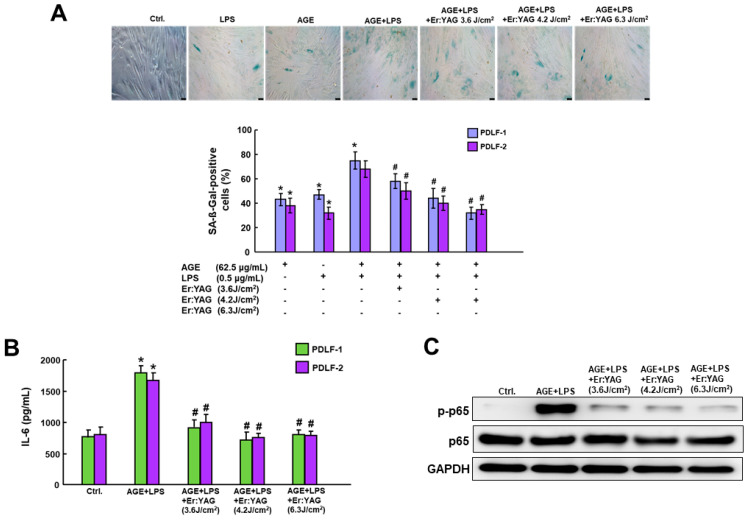
ErL irradiation on cells dose-dependently decreased the AGE/LPS-induced cellular senescence (**A**). ErL intervention significantly suppressed the upregulated IL-6 secretion (**B**) and the phosphorylation activity of p65 in AGEs+LPS group (**C**). * *p* < 0.05 compared to control group; # *p* < 0.05 compared to AGE/LPS only group. Scale bar, 100 µm.

**Figure 4 ijms-25-02116-f004:**
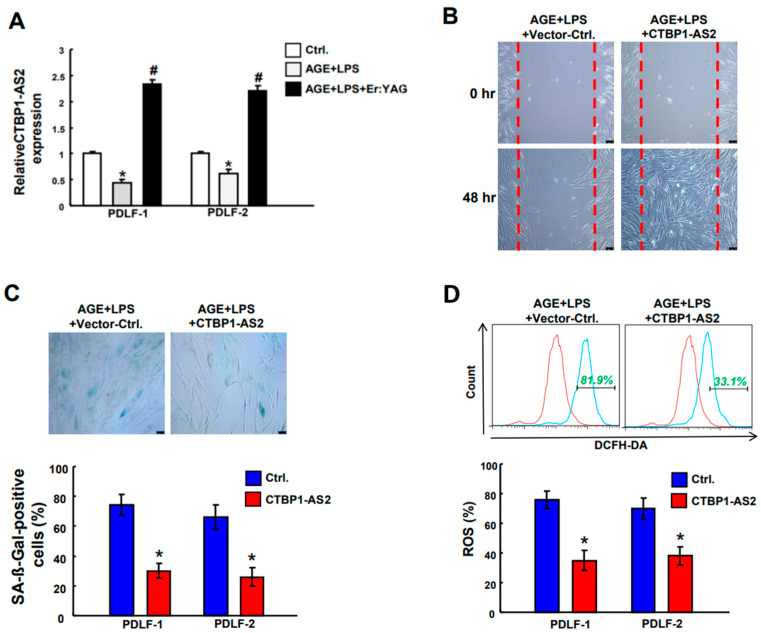
AGE/LPS-stimulated cells had a low expression of CTBP1-AS2 and following ErL intervention, the downregulated lncRNA was restored to a level significantly higher than that of the control group (**A**). The overexpression of CTBP1-AS2 in AGE/LPS-stimulated PDLFs demonstrated significant improvements in wound healing (**B**), cell senescence activity (**C**), and ROS production (**D**). * *p* < 0.05 compared to control group; # *p* < 0.05 compared to AGE/LPS only group. Scale bar, 100 µm.

**Figure 5 ijms-25-02116-f005:**
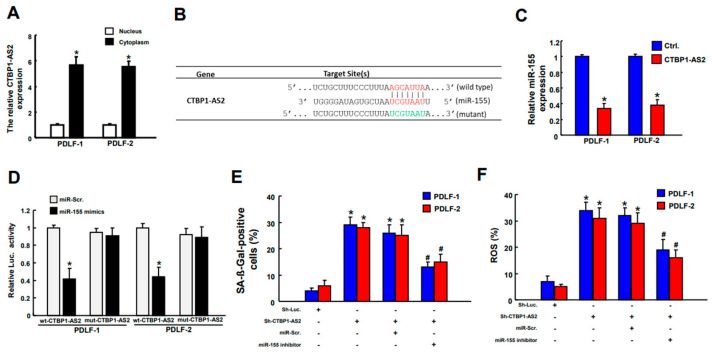
The level of CTBP1-AS2 expression was markedly upregulated in cytoplasm of PDLFs after ErL irradiation (**A**). The illustration of three prime untranslated region (3′ UTR) of CTBP1-AS2 and miR-155 (**B**). Transfection of PDLFs with CTBP1-AS2 resulted in a lower mRNA level of miR-155 (**C**). The luciferase activity of Wt-CTBP1-AS2 vector was reduced when cotransfected with miR-155 mimics, whereas no significant change was present in the mut-CTBP1-AS2 vector in PDLFs (**D**). When miR-155 inhibitor was introduced, the increased cell senescence (**E**) and ROS levels (**F**) were reversed. * *p* < 0.05 compared to control group/miR-Scr. only group; # *p* < 0.05 compared with sh-CTBP1-AS2+miR-Scr. group.

**Figure 6 ijms-25-02116-f006:**
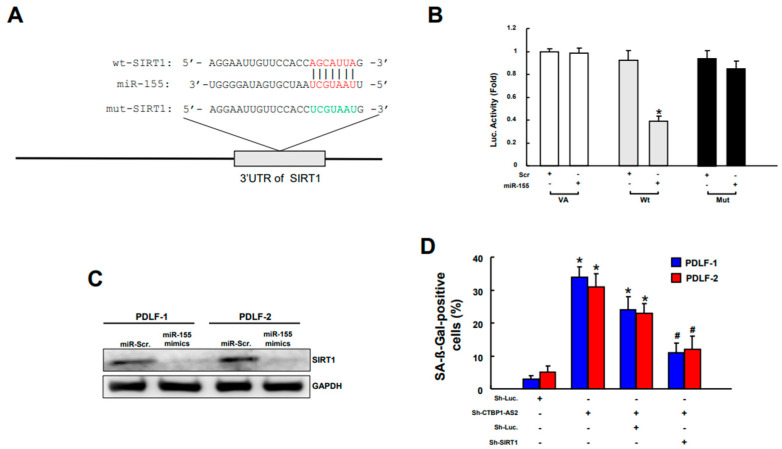
Schematic presentation of the constructed 3′ UTR plasmids of wild-type (Wt) and mutated (Mut) *SIRT1* (**A**). The luciferase activity of each combination was measured and only WT reporter activity was inhibited by miR-155 (**B**). When miR-155 mimics were added to PDLFs, the mRNA level of *SIRT1* decreased significantly (**C**). These data demonstrated that miR-155 negatively regulated *SIRT1* gene expression. The increased cell senescence of CTBP1-AS2-silenced PDLFs was downregulated when *SIRT1* mimics were added (**D**). * *p* < 0.05 compared with miR-Scr. only group; # *p* < 0.05 compared with sh-CTBP1-AS2+miR-Scr. group.

**Figure 7 ijms-25-02116-f007:**
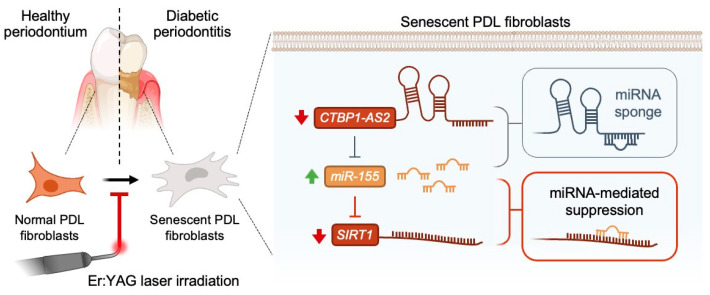
The graphic abstract of the present study illustrates the mechanisms of Er:YAG laser in the regulation of CTBP1-AS2/miR-155/*SIRT1* axis in the amelioration of inflammaging in diabetic periodontitis.

## Data Availability

The data from the current article are available from the corresponding author upon reasonable request.

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
