# Peer review of "Er:YAG Laser Alleviates Inflammaging in Diabetes-Associated Periodontitis via Activation CTBP1-AS2/miR-155/SIRT1 Axis"

_ijms, 2024, doi:10.3390/ijms25042116_

Round 1

Reviewer 1 Report

Comments and Suggestions for Authors

Dear authors,

Thank you for submitting your valuable work to the journal. This in vitro study approaches a significant topic in field of periodontology, that of the diabetic patient with periodontitis. This connection is of major interest to clinicians, given the changes induces by diabetes at periodontal level.

The paper is generally well written and the study's design is sound. However, i would suggests some changes in order to increase its scientific accuracy:

- please provide a Null hypothesis along the objectives of your research

- please add range and sensitivity of the ELISA kit used

- please expand on the statistical tests used (what type of t-test, what type of p - values, one or two-tailed)

- please better explain the formation and composition of each group (test and control)

- please expand on the clinical implications of your results in the Discussion section

- please add separate conclusions to the study

We look forward to receiving the revised version of your manuscript.

Kind regards. 

Comments on the Quality of English Language

moderate editing 

Author Response

Reviewer 1

Comments:
1. Please provide a Null hypothesis along the objectives of your research

Thank you for your comments, we have included a null hypothesis in line 80.

(Line 80) The null hypothesis states that ERL does not exert a significant impact on inflammaging in DP-simulated PDLFs, and it does not modulate the expression of CTBP1-AS2 and its target genes

2. please add range and sensitivity of the ELISA kit used

Thank you for your comments, we have added the information in line 346 (Line 346).

3. Please expand on the statistical tests used (what type of t-test, what type of p - values, one or two-tailed)

Thank you for your recommendations, we have clarified further on the statistical tests used.(Line 393) Study variables were tested for normal distribution by the Shapiro-Wilk test. Intergroup comparisons of means in each parametric and non-parametric parameter were performed using two sample t-test and Wilcoxon rank sum test respectively. Statistical significance was set at two-tailed P values at < 0.05 for all tests.

4.Please better explain the formation and composition of each group (test and control)

Thank you for your comments, we have made modifications accordingly to better explain the test and control groups.(Line 96) As shown in Figure 2, the wound healing was shown the most impaired in PDLFs stimulated with both LPS and AGEs, in comparison to control (no stimuli) or AGEs or LPS stimuli alone.(Line 125) Interestingly, the expression of CTBP1-AS2 in these DP-simulated cells were found tremendously lower compared to that of the control group with no AGEs/LPS stimulus.

5. please expand on the clinical implications of your results in the Discussion section

Thank you for your valuable recommendation, we have further elaborated on the clinical implication of our findings (Line 259) Utilizing this lncRNA as a targeted therapeutic approach could involve strategies to re-store or enhance its expression, aiming to mitigate the progression of periodontal complications in individuals with DM. Additionally, its potential role as a diagnostic marker could offer a non-invasive means of assessing the risk or severity of DP in DM patients, enabling earlier intervention and personalized treatment strategies. Furthermore, considering the increasing association of inflammaging with various inflammatory diseases, this knowledge holds potential to advance therapeutic strategies, addressing heightened host responses not only in DP but also in other DM-related complications, including diabetic nephropathy.

6. Please add separate conclusions to the study

Thank you for your comments, we have added a separate conclusion in Line 274.

Reviewer 2 Report

Comments and Suggestions for Authors

Researchers conducted work on the effect of Erbium-16 doped yttrium aluminum-garnet laser (ErL) in a model of periodontitis and diabetes, this research was focused on the CTBP1-AS2/miR-155/SIRT1 axis. The techniques employed were varied, including cell culture and proliferation, flow cytometry, Western blotting, ELISA, among others.

The results obtained are novel and contribute to clarify the mechanism by which this laser therapy acts, which is relatively new. However, there are some points to be improved. 

1. Throughout the manuscript it is not specified what is meant by PDLF-1 and PDLF-2. Does this refer to each of the patients from whom the samples will be taken? Explain. 

2. In the statistical analysis section, it is not specified if normality tests were applied, and which ones, to decide if it is correct to use a Student's t-test. 

3. I think it is necessary to include a paragraph on the limitations of this study. For example that all these analyses are in vitro.

4. Figure 7 is not mentioned in the text. I suggest that it be moved next to the conclusion and its explanation be extended. 

Author Response

Reviewer 2

1. Throughout the manuscript it is not specified what is meant by PDLF-1 and PDLF-2. Does this refer to each of the patients from whom the samples will be taken? Explain.

Thank you for your valuable comments, we have added an explanation describing what is PDLF-1 and PDLF-2.(Line 96) PDL-1 and PDLF-2 refer to PDLFs obtained from two different individuals

2. In the statistical analysis section, it is not specified if normality tests were applied, and which ones, to decide if it is correct to use a Student’s t-test.

Thank you for pointing this, we have further elaborated on our statistical analysis section.(Line 393) Study variables were tested for normal distribution by the Shapiro-Wilk test. Inter-group comparisons of means in each parametric and non-parametric parameter were performed using two sample t-test and Wilcoxon rank sum test respectively. Statistical significance was set at two-tailed P values at < 0.05 for all tests.

3. I think it is necessary to include a paragraph on the limitations of this study. For example that all these analyses are in vitro.

Thank you for the comments, we have added a paragraph that includes the limitations of this study as well as the clinical implications of the findings in this study.(Line 254) One limitation of this study is that it is solely an in-vitro investigation, and thus, the findings may not fully capture the complexities of DP in vivo. Further animal and pre-clinical studies would be necessary to correlate periodontal tissue loss severity, glycemic control in DM and the studied lncRNA expression. Despite this limitation, the insights gained from this study hold significant promise, enhancing our understanding on  inflammaging in DP through the CTBP1-AS2 pathway. Utilizing this lncRNA as a target-ed therapeutic approach could involve strategies to restore or enhance its expression, aiming to mitigate the progression of periodontal complications in individuals with DM. Additionally, its potential role as a diagnostic marker could offer a non-invasive means of assessing the risk or severity of DP in DM patients, enabling earlier intervention and personalized treatment strategies. Furthermore, considering the increasing association of inflammaging with various inflammatory diseases, this knowledge holds potential to advance therapeutic strategies, addressing heightened host responses not only in DP but also in other DM-related complications, including diabetic nephropathy.

4. Figure 7 is not mentioned in the text. I suggest that it be moved next to the conclusion and its explanation be extended.

Thank you for your recommendation, we have included the elaboration of Fig.7 just before the conclusion.(Line 268) Collectively, these findings highlight the role of CTBP1-AS2 in enhancing SIRT1 ex-pression by inhibiting miR-155 in its amelioration of inflammaging. As illustrated in Fig-ure 7, the diminished expression of CTBP1-AS2 in DP tissues leads to the upregulation of miR-155, subsequently suppressing SIRT1 expression, thereby contributing to inflammaging. The irradiation of ErL, on the other hand, effectively restore the impaired levels of this lncRNA and reversed inflammaging.

Round 2

Reviewer 1 Report

Comments and Suggestions for Authors

Dear authors,

Thank you for submitting the revised version of your paper. The manuscript has improved significantly. I have no further comments.

Kind regards.

Comments on the Quality of English Language

Minor check-up.